# DiffStyle3D: Consistent 3D Gaussian Stylization via Attention Optimization

**Yitong Yang** [* 1]  **Yinglin Wang** [* 1]  **Xuexin Liu** [1]  **Jing Wang** [1]  **Hao Dou** [1]  **Changshuo Wang** [2]  **Shuting He** [1]

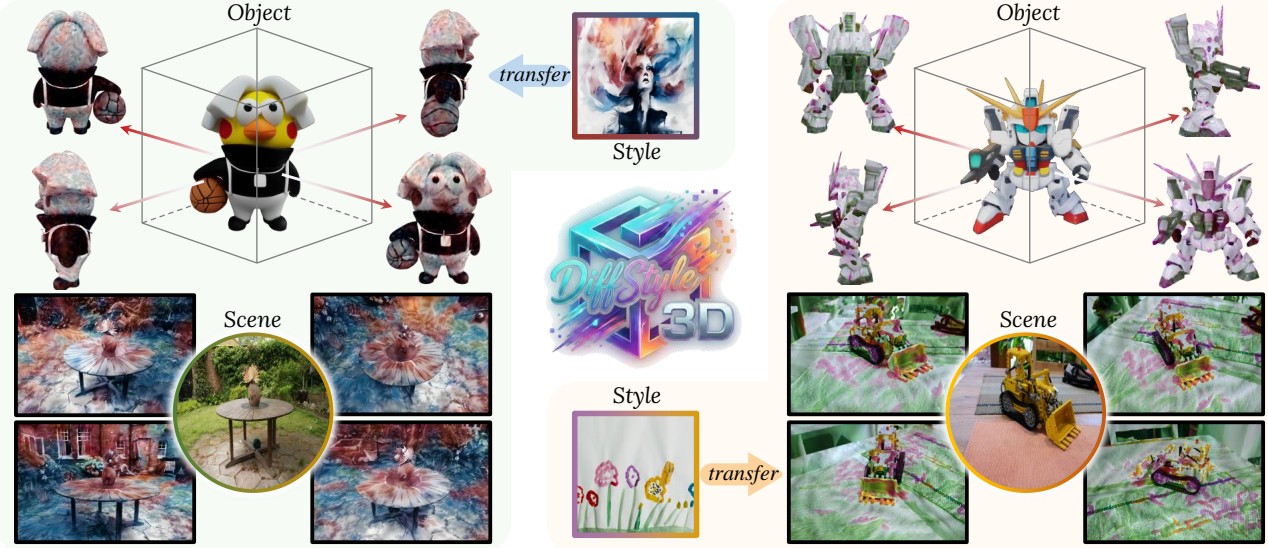

*Figure 1.* Our method enables high-quality 3D stylization across diverse styles for both scenes and objects.

## Abstract

3D style transfer enables the creation of visually expressive 3D content, enriching the visual appearance of 3D scenes and objects. However, existing VGG- and CLIP-based methods struggle to model multi-view consistency within the model itself, while diffusion-based approaches can capture such consistency but rely on denoising directions, leading to unstable training. To address these limitations, we propose DiffStyle3D, a novel diffusion-based paradigm for 3DGS style transfer that directly optimizes in the latent space. Specifically, we introduce an Attention-Aware Loss that performs style transfer by aligning style features in the self-attention space, while preserving original content through content feature alignment. Inspired by the geometric invariance of 3D

stylization, we propose a Geometry-Guided Multi-View Consistency method that integrates geometric information into self-attention to enable cross-view correspondence modeling. Based on geometric information, we additionally construct a geometry-aware mask to prevent redundant optimization in overlapping regions across views, which further improves multi-view consistency. Extensive experiments show that DiffStyle3D outperforms state-of-the-art methods, achieving higher stylization quality and visual realism. The code is available at https://github.com/yangyt46/DiffStyle3D.

## 1. Introduction

With the rapid development of applications such as virtual reality, gaming, and film production, the demand for digital content is shifting from 2D images to 3D representations, making large-scale, high-quality 3D assets increasingly essential (He et al., 2025; Wang et al., 2026). Against this backdrop, 3D stylization has emerged as a promising research direction, aiming to transform static 3D geometric representations into expressive digital assets with distinctive aesthetic characteristics, thereby facilitating the low-cost, efficient, and scalable creation of high-quality 3D artistic

---

[*]Equal contribution [1]School of Computing and Artificial Intelligence, Shanghai University of Finance and Economics, Shanghai, China [2]Department of Computer Science University College London, London, United Kingdom. Correspondence to: Yinglin Wang <wang.yinglin@shufe.edu.cn>, Shuting He <shuting.he@sufe.edu.cn>.

*Proceedings of the $43^{rd}$ International Conference on Machine Learning*, Seoul, South Korea. PMLR 306, 2026. Copyright 2026 by the author(s).

content. Previous 3D style transfer methods (Chen et al., 2024b; Liu et al., 2023a; Fujiwara et al., 2024; Zhang et al., 2022) predominantly relied on NeRF-based (Mildenhall et al., 2021) representations, which suffer from substantial computational overhead and long training times, limiting their efficiency and scalability. Recently, 3D Gaussian Splatting (3DGS) (Kerbl et al., 2023) has emerged as a promising alternative, offering significantly improved rendering efficiency and high visual quality. As a result, 3DGS has quickly become a focal point in 3D style transfer research.

Currently, 3DGS-based style transfer research (Zhuang et al., 2025; Gu et al., 2024; Lin et al., 2025; Zhang et al., 2024) can be broadly divided into three categories. First, VGG-based methods (Liu et al., 2024; Galerne et al., 2025; Lin et al., 2025; Saroha et al., 2024), inspired by 2D feature statistic matching (Gatys et al., 2016; Jing et al., 2019), enforce style consistency by minimizing Gram matrix discrepancies. While these methods offer stable training, inherent model limitations hinder multi-view consistency, often resulting in inconsistent stylization across varying viewpoints. Second, CLIP-based methods (Howil et al., 2025; Kovács et al., 2024) align feature directions within the CLIP embedding space to introduce semantic style constraints; however, they similarly fail to explicitly model cross-view correspondences, leading to stylistic drift or flickering. Recently, diffusion-based methods (Yang et al., 2026) leverage the intrinsic properties of diffusion models to establish multi-view consistency. Nevertheless, these approaches frequently suffer from unstable training and potential artifacts, such as over-smoothing, due to their reliance on optimization along predicted denoising directions.

To address these challenges, we propose DiffStyle3D, a novel diffusion-based paradigm for 3DGS stylization. Unlike previous approaches that rely on denoising directions from diffusion models for optimization, we introduce an Attention-Aware Loss that achieves stable 3D style transfer through direct latent-space optimization while effectively preserving the original content. It consists of two terms. (1) Style loss: within self-attention, we inject the keys (K) and values (V) from the style image into the original queries (Q), using the resulting attention output as a stylization signal to guide the integration of style information into the 3D representation. (2) Content loss: to preserve content fidelity, we align the attention outputs of the content image with those of the rendered image.

Motivated by the geometric invariance of 3D stylization, where only color-related parameters are optimized while geometry remains fixed, we propose Geometry-Guided Multi-View Consistency. By leveraging camera parameters and depth maps, we explicitly determine geometric relationships and incorporate this information into the self-attention mechanism to form Geometry-Guided Attention, thereby

modeling cross-view correspondences and mitigating view conflicts caused by inconsistent style information. Additionally, based on the geometric information, we introduce a geometry-aware mask to prevent redundant optimization in multi-view overlapping regions, further improving multi-view consistency. In summary, our key contributions are as follows:

- To the best of our knowledge, DiffStyle3D is the first paradigm to perform 3DGS stylization by optimizing directly in the latent space of a diffusion model.

- We propose an Attention-Aware Loss that enables style transfer while preserving the original content.

- We propose Geometry-Guided Multi-View Consistency to mitigate multi-view inconsistency.

- Extensive experiments demonstrate that our method outperforms existing state-of-the-art approaches in both qualitative and quantitative evaluations.

## 2. Related Work

**2D Style Transfer**. Style transfer has remained a central topic in generative visual research, aiming to map the stylistic characteristics of a reference image onto a content image. The early pioneering approach (Gatys et al., 2016) achieved neural style transfer by minimizing the distance between Gram matrices derived from VGG features, which motivated extensive follow-up works (Heitz et al., 2021; Risser et al., 2017; Vacher et al., 2020). With the rapid development of diffusion models, existing methods have increasingly relied on this framework, leading to the emergence of numerous fine-tuning-based approaches (Zhou et al., 2025; Ye et al., 2023; Xing et al., 2024; Yang et al., 2025a) and training-free methods (He et al., 2024; Xu et al., 2024; Wang et al., 2024). Compared to traditional VGG-feature-based methods, diffusion-driven approaches achieve higher content fidelity and improved stylistic expressiveness. Therefore, we propose DiffStyle3D, a fully diffusion-based framework.

**Attention Control**. As one of the most advanced paradigms in generative modeling, diffusion models (Ho et al., 2020; Nichol & Dhariwal, 2021; Rombach et al., 2022) have demonstrated remarkable capabilities across both 2D and 3D domains (Podell et al., 2023; Liu et al., 2023b; Shi et al., 2023). At the heart of these models lies the attention mechanism, which has been extensively explored in recent research (Alaluf et al., 2024; Cao et al., 2023). By imposing various forms of control on self-attention and cross-attention modules, existing methods have achieved superior performance in tasks such as content editing (Yang et al., 2025b; Hertz et al., 2022) and style transfer (Hertz et al., 2024; Chung et al., 2024). Based on the self-attention mechanism,

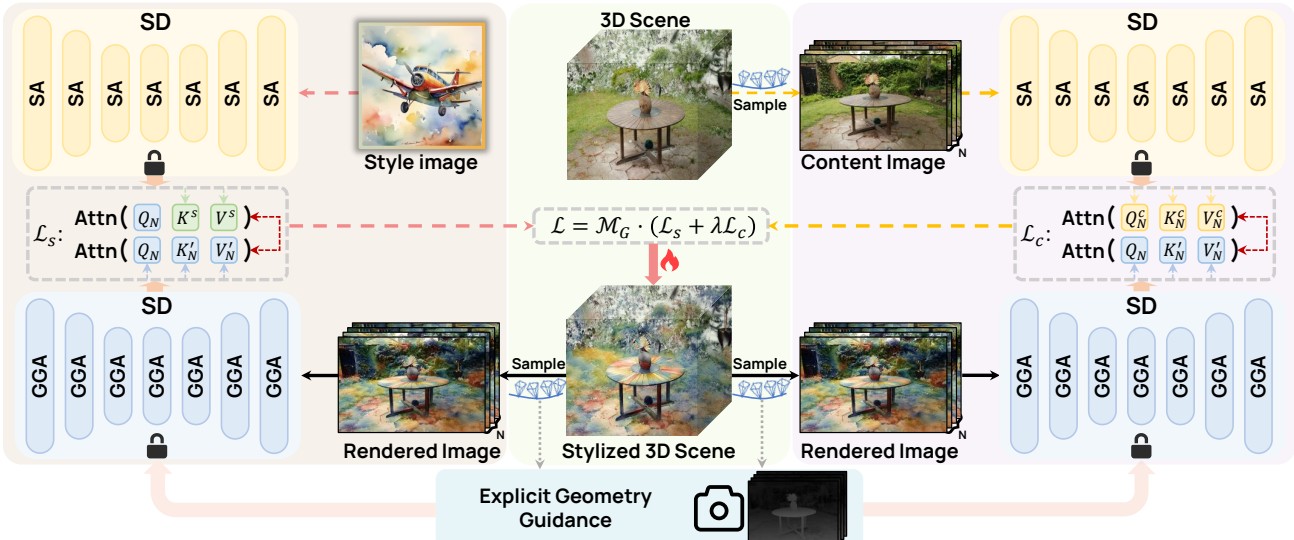

*Figure 2.* Overview of DiffStyle3D. We introduce an Attention-Aware Loss that enables style transfer while preserving content. To model multi-view correspondences, we derive explicit geometric guidance from camera parameters and depth maps, which is then incorporated into Self-Attention (SA) to form Geometry-Guided Attention (GGA). Additionally, a geometry-aware mask $\mathcal{M}_G$ restricts optimization to non-overlapping regions, further improving multi-view consistency.

we propose an Attention-Aware Loss to effectively transfer stylistic information while preserving the original content.

**3DGS Style Transfer**. Following a trajectory similar to that of 2D style transfer, style transfer methods for 3D Gaussian Splatting can be broadly categorized into three groups. Most existing approaches are VGG-based methods, which optimize the 3D scene using feature matching losses (Liu et al., 2024; Saroha et al., 2024), multi-scale losses (Galerne et al., 2025), or nearest-neighbor feature matching losses (Jain et al., 2024; Zhang et al., 2024). Another line of CLIP-based methods (Howil et al., 2025; Kovács et al., 2024) achieves stylization by aligning representations in the CLIP embedding space. More recently, diffusion-based methods (Zhuang et al., 2025) have been explored for 3D stylization, either by generating stylized 2D image supervision using diffusion models (Gu et al., 2024; Yu et al., 2024), which essentially reformulates the problem as a 2D style transfer task, or by directly distilling diffusion models into the 3D representation (Yang et al., 2026). However, these approaches struggle to effectively establish multi-view consistency. In contrast, we introduce Geometry-Guided Multi-View Consistency to enforce cross-view consistency.

## 3. Preliminary

**Self-Attention.** Given an input sequence $X \in \mathbb{R}^{n \times d}$, the self-attention mechanism projects each token into a triplet of latent representations consisting of query $Q$, key $K$, and value $V$ through learned linear transformations. The attention output is computed as

$$\text{Attn}(Q, K, V) = \text{Softmax}\left(\frac{QK^\top}{\sqrt{d_k}}\right) V, \quad (1)$$

where $d_k$ is the scaling factor. This mechanism allows each token to aggregate information from others according to pairwise relevance, capturing contextual and content-dependent interactions.

**3D Gaussian Splatting.** 3DGS represents a 3D scene as a set of $M$ anisotropic Gaussians, formulated as:

$$\min_{\Theta} \frac{1}{N} \sum_{i=1}^{N} \mathcal{L}(\mathcal{R}(C_i; \Theta), I_i^{gt}), \quad (2)$$

where $\Theta = \{(\mu_m, \Sigma_m, \alpha_m, \mathcal{C}_m)\}_{m=1}^{M}$ denotes the 3D Gaussian parameters. Here, $\mu_m, \Sigma_m, \alpha_m$, and $\mathcal{C}_m$ represent the mean position, covariance matrix, opacity, and spherical harmonics (SH) coefficients for color, respectively. $C_i$ denotes the $i$-th camera parameters, $\mathcal{R}(\cdot)$ the rasterization-based renderer, and $I_i^{gt}$ the corresponding ground-truth image.

## 4. Method

Given a style image $I^s$, we aim to transfer its style to the 3D scene while preserving the original content. Therefore, we only optimize the color-related parameters. We propose DiffStyle3D (Fig. 2), which employs an Attention-Aware Loss for 3D style transfer via direct latent-space optimization (Sec. 4.1). To capture cross-view correspondences and improve multi-view consistency, we introduce Geometry-Guided Multi-View Consistency (Sec. 4.2).

### 4.1. Attention-Aware Loss

Previous diffusion model–based approaches for 3D style transfer (Yang et al., 2026; Zhuang et al., 2025) typically optimize 3D representations by following the predicted de-

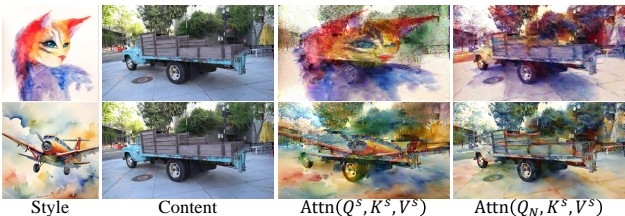

*Figure 3.* Results with different stylization signals. We conduct experiments using a fixed viewpoint of the 3D scene. Directly using the attention outputs of the style image as stylization signals leads to severe content leakage.

noising directions. However, this optimization strategy often leads to instability and overly smooth results, limiting their effectiveness in high-quality 3D stylization. Drawing inspiration from inference-time optimization techniques (Chen et al., 2024a; Ding et al., 2024; Shi et al., 2024), we introduce an Attention-Aware Loss that establishes a new 3D stylization paradigm through direct optimization in the latent space of diffusion models. By aligning representations in the self-attention feature space, our method enables accurate style transfer while preserving original content, without relying on unstable denoising guidance.

Specifically, given a 3D Gaussian scene and a style image $I^s$, we sample $N$ cameras $C_N = \{c_1, c_2, \ldots, c_N\}$ in each batch to render the scene, producing a set of rendered images $I_N$ along with their corresponding original content images $I_N^c$. These images are then fed into a diffusion model to extract features, which can be formulated as follows:

$$z_N = E(I_N), \quad z_N^c = E(I_N^c), \quad z^s = E(I^s),$$
$$h_N := \epsilon_\theta(z_N), \quad h_N^c := \epsilon_\theta(z_N^c), \quad h^s := \epsilon_\theta(z^s), \quad (3)$$

where $E(\cdot)$ denotes the VAE encoder, $\epsilon_\theta$ represents the UNet. $h_N$, $h_N^c$, and $h^s$ denote the features extracted from specific layers of the UNet, which are used to compute the style and content losses.

**Style Loss.** Controlling self-attention has been widely adopted in style transfer tasks, motivating us to design our loss function around the self-attention mechanism. A straightforward solution is to directly align the attention outputs of the style image and the rendered image to achieve style transfer. However, such a strategy often leads to severe content leakage from the style image, as illustrated in Fig. 3. To address this issue, we inject stylistic semantics by combining the key (K) and value (V) from the style image with the query (Q) from the rendered image and use the resulting attention output as the stylization signal. Formally, we first extract the $Q_N$, $K_N$, and $V_N$ from the self-attention layers of the rendered image to compute the attention output, which are centered to zero mean and then normalized:

$$\widehat{\mathcal{A}}_N = \frac{\mathcal{A}_N - \mu(\mathcal{A}_N)}{\|\mathcal{A}_N - \mu(\mathcal{A}_N)\|_2}, \mathcal{A}_N = \text{Attn}(Q_N, K_N, V_N), \quad (4)$$

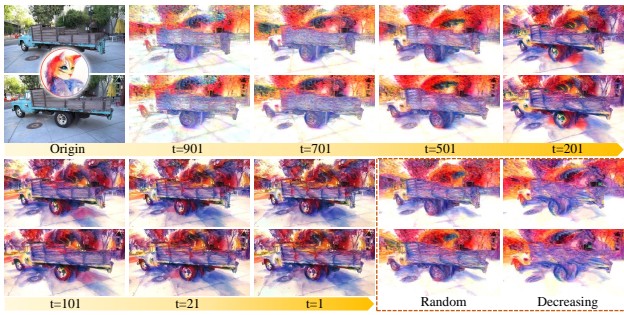

*Figure 4.* Results obtained using different timestep during optimization. *Random* denotes randomly sampled timestep throughout the optimization process, while *decreasing* simulates the diffusion process by progressively decreasing the time step from $T$ to 0.

where $\mu(\cdot)$ denotes the mean over channels. Meanwhile, $K^s$ and $V^s$ from the style image are integrated with $Q_N$ to inject style semantics:

$$\widehat{\mathcal{A}}^s = \frac{\mathcal{A}^s - \mu(\mathcal{A}^s)}{\|\mathcal{A}^s - \mu(\mathcal{A}^s)\|_2}, \mathcal{A}^s = \text{Attn}(Q_N, K^s, V^s). \quad (5)$$

Finally, style guidance is achieved by minimizing the distance between the two representations:

$$\mathcal{L}_s = \left\|\widehat{\mathcal{A}}_N - \widehat{\mathcal{A}}^s\right\|_2^2. \quad (6)$$

Directly applying $\ell_1$ or $\ell_2$ loss on $\mathcal{A}^s$ and $\mathcal{A}_N$ focuses on the absolute values of the features, which can slow down training. By centering and normalizing features before computing the loss, we emphasize the consistency of their direction and patterns, enabling a more effective style transfer.

**Content Loss.** A central challenge in style transfer is to effectively apply the target style while preserving the original content. To prevent over-stylization that could distort local features or semantic structures in the rendered image, we design a content loss. Similar to the style loss, the content loss is defined based on self-attention. It preserves the original content by minimizing the distance between the attention representations of the content image and the rendered image. Formally, it can be expressed as follows:

$$\widehat{\mathcal{A}}_N^c = \frac{\mathcal{A}_N^c - \mu(\mathcal{A}_N^c)}{\|\mathcal{A}_N^c - \mu(\mathcal{A}_N^c)\|_2}, \mathcal{A}_N^c = \text{Attn}(Q_N^c, K_N^c, V_N^c),$$
$$(7)$$
$$\mathcal{L}_c = \left\|\widehat{\mathcal{A}}_N - \widehat{\mathcal{A}}_N^c\right\|_2^2. \quad (8)$$

**Timestep Choice.** As illustrated in Fig. 4, we analyze the effect of different fixed diffusion timesteps during optimization. Larger timesteps introduce increased noise and result in blurred stylization, while smaller timesteps better preserve fine-grained stylistic details and brushstroke textures. We further consider *random* and *decreasing* timestep strategies commonly used in 3D generation. However, since these strategies still involve large timesteps, they tend to introduce

blurring artifacts. As a result, we adopt a fixed timestep of $t = 1$ as a key choice in our method.

## 4.2. Geometry-Guided Multi-View Consistency

Although the Attention-Aware Loss achieves promising results in style transfer, the same object may still receive inconsistent stylistic representations across different views, leading to noticeable cross-view inconsistency. To address this issue, we propose Geometry-Guided Multi-View Consistency. Unlike VGG-based (Galerne et al., 2025) and CLIP-based methods (Howil et al., 2025), which struggle to model such cross-view constraints within the model, our approach leverages the intrinsic self-attention mechanism of diffusion models to capture correlations across different views, thereby improving multi-view consistency.

**Explicit Geometry Guidance**. In our framework, we optimize only color-related parameters, ensuring geometric invariance of the 3D Gaussians and a fixed depth map $D_b$ for any given viewpoint $b$. This stability allows us to explicitly establish geometric correspondences across views using known camera intrinsics and poses. For a pixel $\mathbf{p}$ in the reference view $b$, its corresponding sampling coordinate in source view $j$, denoted as $\mathbf{g}_{b \leftarrow j}(\mathbf{p})$, is derived via back-projection and re-projection:

$$\mathbf{g}_{b \leftarrow j}(\mathbf{p}) = \Pi\left(\mathbf{K}_j \, \mathbf{T}_j^{w2c} \mathbf{T}_b^{c2w} D_b(\mathbf{p}) \mathbf{K}_b^{-1} \tilde{\mathbf{p}}\right), \quad (9)$$

where $\tilde{\mathbf{p}}$ represents the homogeneous coordinates of $\mathbf{p}$, while $\mathbf{K}$ and $\mathbf{T}$ denote camera intrinsics and extrinsics. $w2c$ and $c2w$ denote the world-to-camera and camera-to-world transformations, respectively. $\Pi(\cdot)$ represents perspective projection followed by normalization to the $[-1, 1]$ sampling space. To account for occlusions and boundaries, we define a visibility mask:

$$\mathbf{v}_{b \leftarrow j}(\mathbf{p}) = \mathbf{1}\left(\text{inFront}_j(\mathbf{p}) \, \wedge \, \mathbf{g}_{b \leftarrow j}(\mathbf{p}) \in \Omega\right), \quad (10)$$

where $\mathbf{1}(\cdot)$ denotes the indicator function, $\Omega$ denotes the valid image domain, and $\text{inFront}_j(\cdot)$ enforces that the reprojected point has a positive depth in view $j$.

**Geometry-Guided Attention.** We integrate the obtained sampling grids and visibility masks into all self-attention layers of the diffusion model to explicitly model correspondences across multiple views, thereby strengthening cross-view consistency constraints. Specifically, we augment $K$ and $V$ of the reference view by warping features from all other views within the batch. For a batch of $N$ views, $K_b'$ and $V_b'$ for view $b$ are formulated as:

$$K_b' = [K_b; \, \{\mathcal{W}_{b \leftarrow j}(K_j) \mid j \in \{0, \dots, N-1\}, \, j \neq b\}],$$
$$V_b' = [V_b; \, \{\mathcal{W}_{b \leftarrow j}(V_j) \mid j \in \{0, \dots, N-1\}, \, j \neq b\}], \quad (11)$$

where $[\cdot \, ; \, \cdot]$ denotes the concatenation and $\mathcal{W}_{b \leftarrow j}(\cdot)$ represents the bilinear warping operator guided by $\mathbf{g}_{b \leftarrow j}$. By

rewriting $\mathcal{A}_N$ in Eq. 4, the Geometry-Guided Attention (GGA) formula is defined as:

$$\text{Attn}(Q_N, K_N', V_N') = \text{Softmax}\left(\frac{Q_N K_N'^\top}{\sqrt{d_k}} + \mathbf{M_v}\right) V_N', \quad (12)$$

where $K_N' = \{K_b'\}_{b=0}^{N-1}$, $V_N' = \{V_b'\}_{b=0}^{N-1}$. $\mathbf{M_v}$ consists of visibility masks $\mathbf{v}$ and serves as the attention mask to prevent erroneous feature aggregation from occluded regions.

**Geometry-Aware Mask.** To avoid redundant optimization over multi-view overlapping regions, we introduce a geometry-aware mask $\mathcal{M}_G$, which is defined as follows:

$$\mathcal{M}_G \coloneqq \{\mathbf{1}(\forall j < b, \, \mathbf{v}_{b \leftarrow j}(\mathbf{p}) = 0)\}_{b, \mathbf{p}}. \quad (13)$$

$\mathcal{M}_G$ is a collection of masks, one for each view. For a given view $b$, its mask retains only the pixels that have not been observed in any previous view $j < b$, setting them to 1. Consequently, $\mathcal{M}_G$ represents the non-overlapping regions across all views in the current batch, which further improves multi-view consistency.

**Optimization Objective.** We extract features from all self-attention layers of the diffusion model. The final loss is defined as follows:

$$\mathcal{L} = \mathcal{M}_G \cdot (\mathcal{L}_s + \lambda \mathcal{L}_c), \quad (14)$$

where $\lambda$ is a scaling factor that controls the strength of content preservation during style transfer. Our method is detailed in Alg. 1.

---

**Algorithm 1** DiffStyle3D

---

**Require:** VAE encoder $E$, diffusion model $\epsilon_\theta$, GGA-integrated diffusion model $\epsilon_\theta'$, training iterations $S$, fixed timestep $t$, $N$ views per batch, style image $I^s$.
1: **for** $s = 1$ to $S$ **do**
2:   **Sample:** $I_N = \mathcal{R}(C_N; \Theta)$, $I_N^c$, depth maps $D_N$, camera $C_N$ intrinsics and extrinsics.
3:   **Explicit Geometry Guidance:** $\mathbf{g}$ and $\mathbf{v}$ defined by Eq. 9, Eq. 10, geometry-aware mask $\mathbf{M}_G$.
4:   $z_N, \, z_N^c, \, z^s \leftarrow E(I_N), \, E(I_N^c), \, E(I^s)$
    $\widehat{\mathcal{A}}_N^c, Q_N \leftarrow h_N^c \coloneqq \epsilon_\theta(z_N^c, t)$,     in Eq. 7
5:   $\widehat{\mathcal{A}}^s \leftarrow \{Q_N, \quad h^s \coloneqq \epsilon_\theta(z^s, t)\}$,     in Eq. 5
    $\widehat{\mathcal{A}}_N \leftarrow h_N \coloneqq \epsilon_\theta'(z_N, t)$,     in Eq. 4, 11, 12
6:   $\mathcal{L} = \mathcal{M}_G(\mathcal{L}_s(\widehat{\mathcal{A}}_N, \widehat{\mathcal{A}}^s) + \lambda \mathcal{L}_c(\widehat{\mathcal{A}}_N, \widehat{\mathcal{A}}_N^c))$ Eq. 6, 8
7:   Compute $\nabla_{z_N} \mathcal{L}$
8:   Update 3D Gaussians using $\nabla_{z_N} \mathcal{L}$
9: **end for**
**Ensure:** 3D Stylized Scene.

---

*Table 1.* Quantitative comparison of different methods in 3DGS style transfer. **Bold** : best; underline : second best.

| Method | CLIP-S ↑ | CLIP-C ↑ | CLIP-CONS ↑ | CLIP-F | $S_{vgg}$ ↓ | FID ↓ | Short-range consistency | | Long-range consistency | | Per-Instance training time↓ |
| | | | | | | | LPIPS ↓ | RMSE ↓ | LPIPS ↓ | RMSE ↓ | |
| --- | --- | --- | --- | --- | --- | --- | --- | --- | --- | --- | --- |
| Scene-level | | | | | | | | | | | |
| StyleGaussian | 0.64 | 0.61 | 0.032 | 1.02 | 25.81 | 334.9 | 0.088 | 0.110 | 0.151 | 0.160 | ∼21min |
| SGSST | 0.65 | 0.64 | 0.066 | 1.01 | **20.60** | 289.1 | 0.086 | 0.107 | 0.159 | 0.182 | ∼40min |
| FantasyStyle | 0.64 | 0.67 | 0.084 | 1.01 | 26.88 | 228.6 | 0.084 | 0.101 | 0.167 | 0.173 | ∼34min |
| CLIPGaussian | **0.79** | 0.63 | 0.058 | 1.03 | 24.68 | 280.7 | 0.081 | **0.074** | 0.161 | **0.127** | ≃16min |
| Ours | 0.65 | **0.71** | **0.128** | **1.00** | 20.63 | **204.8** | **0.075** | 0.075 | **0.141** | 0.130 | **∼16min** |
| Object-level | | | | | | | | | | | |
| SGSST | 0.63 | 0.75 | 0.224 | 1.02 | **13.94** | 333.7 | 0.184 | 0.176 | 0.211 | 0.202 | ∼6min |
| FantasyStyle | 0.61 | 0.79 | 0.358 | 1.02 | 18.20 | 265.2 | 0.186 | 0.150 | 0.218 | 0.188 | ∼5min |
| CLIPGaussian | **0.72** | 0.79 | **0.504** | 1.03 | 16.70 | 259.6 | 0.177 | **0.111** | 0.209 | **0.155** | ∼13min |
| Ours | 0.63 | **0.84** | 0.412 | **1.01** | 16.44 | **255.8** | 0.176 | 0.119 | **0.208** | 0.161 | **∼4min** |

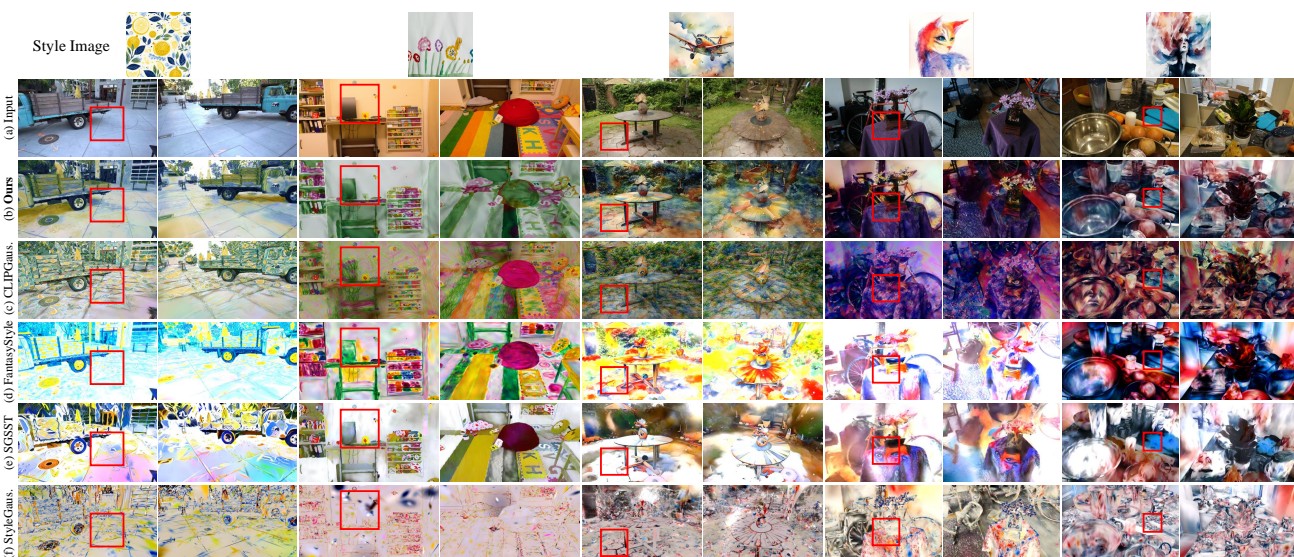

*Figure 5.* Qualitative comparison of different methods on scene-level datasets. Our approach achieves superior style transfer while better preserving the original content. The red boxes highlight clear differences, the details of which are further compared in Fig. 7.

# 5. Experiments

**Datasets.** We select 8 scenes from the Tandt DB dataset (Kerbl et al., 2023) and the Mip-NeRF 360 dataset (Barron et al., 2022). Each scene is stylized using 14 different style images, resulting in a total of 112 stylization experiments. In addition, we employ SAM3D (Chen et al., 2025) to extract 10 individual objects, producing 140 object-level stylization results in total. We comprehensively evaluate our method at both the scene-level and the object-level across diverse artistic domains.

**Metrics.** We evaluate content preservation using CLIP-C (Radford et al., 2021) and FID, while CLIP-S is used to assess style transfer quality. CLIP-CONS and CLIP-F (Howil et al., 2025) are employed to measure semantic temporal consistency, where CLIP-F values closer to 1 indicate better consistency. To evaluate overall transfer quality, we define the $S_{vgg}$, which is computed using features extracted from VGG19 (Simonyan & Zisserman, 2014). Furthermore, LPIPS and RMSE are used to measure short-term and long-term multi-view consistency (Liu et al., 2024), respectively.

Evaluation metric details are in the Appendix B.

**Comparison Methods.** We compare our method with recent state-of-the-art approaches, including VGG-based methods (StyleGaussian (Liu et al., 2024), SGSST (Galerne et al., 2025)), CLIP-based methods (CLIPGaussian (Howil et al., 2025)), and diffusion-based methods (FantasyStyle (Yang et al., 2026)).

**Implementation Details**. We adopt Stable Diffusion 1.5 (Rombach et al., 2022) as our base model. We fix the timestep to $t = 1$ and extract self-attention features from all blocks for loss computation. For each batch, we use $N = 4$ views and set $\lambda = 0.1$. All experiments are conducted on a single NVIDIA L20 (48G) GPU.

## 5.1. Comparison Results

**Quantitative Comparisons**. As shown in Tab. 1, we comprehensively evaluate our method across multiple metrics covering style transfer quality, content preservation, and multi-view consistency. Overall, our method achieves the best performance across these aspects. Specifically, CLIP-

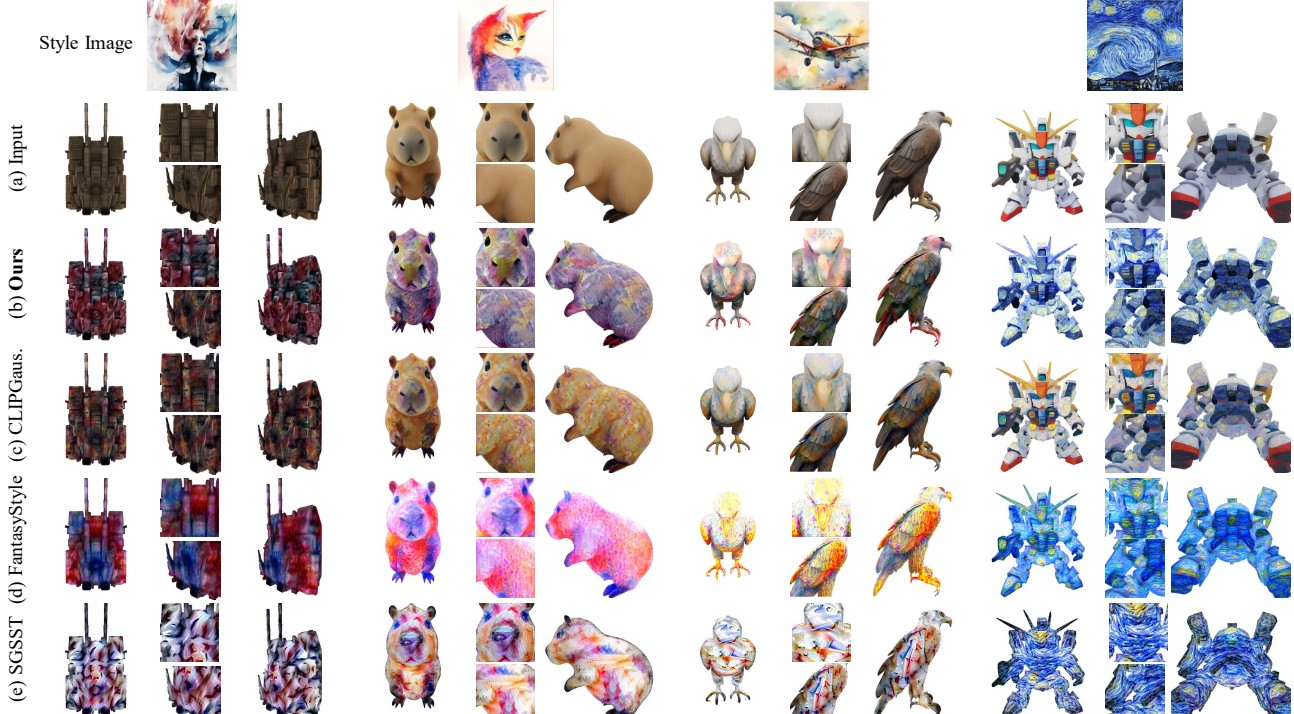

*Figure 6.* Qualitative comparison of different methods on object-level datasets. Other methods often suffer from over-stylization and content leakage from the style image. In contrast, our approach avoids these issues, achieving superior visual quality in style transfer.

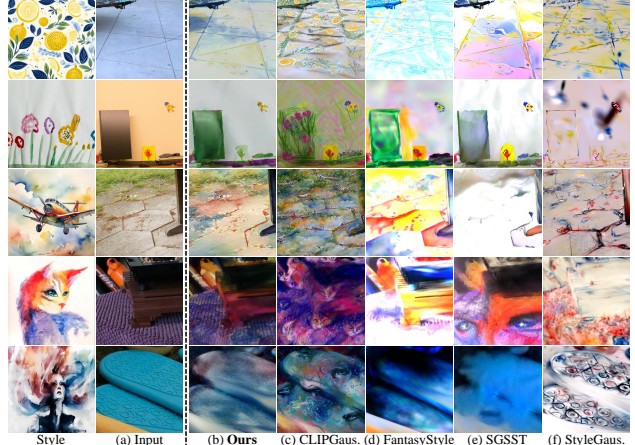

*Figure 7.* Detailed qualitative comparison of different methods. Our approach adheres more closely to the target style and avoids content leakage from the style image, compared to existing methods. Zoom in for better view.

Gaussian attains very high scores on CLIP-S, as it directly optimizes style transfer using CLIP-extracted features; however, this comes at the cost of inferior content preservation. Similarly, SGSST performs well on the $S_{vgg}$ metric due to explicitly optimizing this objective. Despite not being tailored to any single metric, our method achieves competitive or superior performance across all style and content metrics. Notably, our method demonstrates significant improvements on multi-view consistency metrics, such as CLIP-CONS and LPIPS, highlighting its effectiveness in enforcing cross-view

*Table 2.* User study on 3DGS style transfer methods.

| Method | Style Gaussian | SGSST | Fantasy Style | CLIP Gaussian | Ours |
|---|---|---|---|---|---|
| Rank 1(%)↑ | 7.26 | 28.08 | 17.26 | 4.11 | **43.29** |

coherence. In addition, we compare the training time of different methods. Although diffusion models are substantially larger than CLIP and VGG, our method achieves training time comparable to CLIPGaussian and substantially outperforms other methods.

**Qualitative Comparisons**. Fig. 5, 6, and 7 present visual comparisons of different methods on both scene-level and object-level datasets. It can be clearly observed that StyleGaussian struggles to faithfully transfer the target style and severely damages the original scene content. SGSST, based on VGG features, suffers from VGG's limited representational capacity, making it difficult to handle complex style images and resulting in unsatisfactory stylization (e.g., 5th and 6th columns in Fig. 5). It can also produce noticeable color distortions, such as large pink and blue regions on the ground (1st and 2nd columns), and may cause content leakage from the style image (4th row in Fig. 7) or overstylization that obscures the original content (5th row in Fig. 7). FantasyStyle is able to preserve the original content relatively well; however, as it relies on IP-Adapter for style transfer, it fails to fully align with the target style. As a result, some stylized outputs deviate from the intended style appearance (e.g., the 1st, 2nd, 5th, and 6th columns

*Table 3.* Quantitative results of the ablation study on the effect of Geometry-Guided Multi-View Consistency.

| Method | CLIP-CONS↑ | CLIP-F | Short-range Consistency | | Long-range Consistency | |
|---|---|---|---|---|---|---|
| | | | LPIPS ↓ | RMSE ↓ | LPIPS ↓ | RMSE ↓ |
| w/o GGA | 0.121 | 1.01 | 0.078 | 0.079 | 0.151 | 0.140 |
| w/o $\mathcal{M}_G$ | 0.124 | 1.01 | 0.076 | 0.076 | 0.142 | 0.132 |
| Ours | **0.128** | **1.00** | **0.075** | **0.075** | **0.141** | **0.130** |

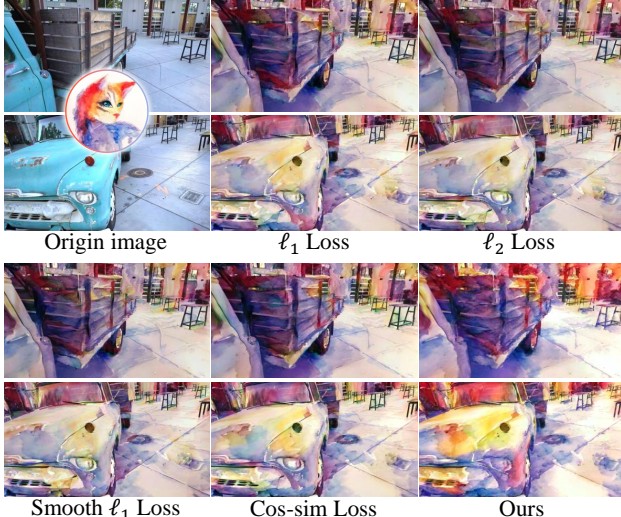

*Figure 8.* Results of directly applying the losses defined in Eq. 4, 5 and 7 without centering and normalization. In contrast, our method achieves faster style transfer.

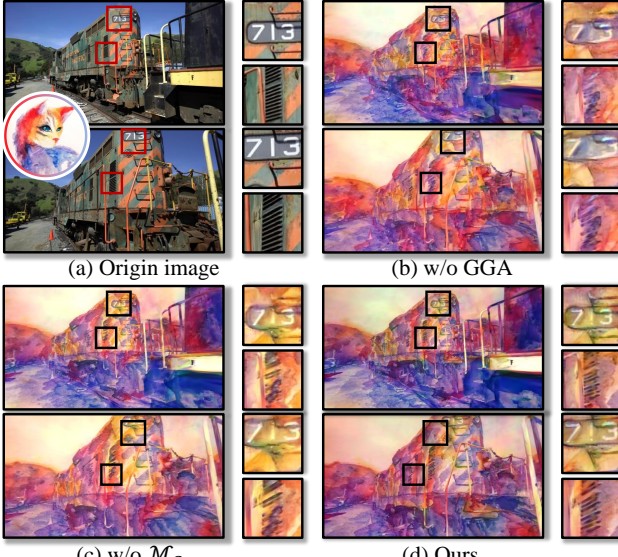

*Figure 9.* Qualitative results of the ablation study on Geometry-Guided Multi-View Consistency. Zoom in for better view.

in Fig. 5). CLIPGaussian, which performs stylization by aligning CLIP features, suffers from severe content leakage from the style image, as shown in Fig. 7 and Fig. 6. It often introduces explicit semantic elements from the style image, such as cat faces or human eyes, which also explains its superior performance on the CLIP-S metric. In contrast, our approach aligns with the target style more accurately than diffusion-based methods, yielding higher-quality style transfer results. Compared with VGG- and CLIP-based methods, it better preserves the original content and effectively avoids content leakage from the style image. Overall, our method achieves the best visual quality among all compared approaches. Additional results are in Appendix D.

**User study**. As shown in Table 2, we conduct a user study to evaluate both stylistic fidelity and content preservation. We collect a total of 730 votes from 73 participants, where each participant is shown ten randomly selected instances and asked to choose the result that best satisfies the evaluation criteria. Our approach significantly outperforms the baseline methods, receiving 43.29% of the overall user preference, thereby demonstrating its effectiveness.

### 5.2. Ablation Study

**Attention-Aware Loss.** We conduct an ablation study on the centering and normalization operations in the Attention-Aware Loss under the same optimization settings, with the

results shown in Fig. 8. Without centering and normalization, style transfer remains incomplete, as the optimization process overemphasizes absolute feature magnitudes, resulting in slow convergence. In contrast, incorporating centering and normalization shifts the optimization focus toward feature directions, enabling faster convergence and more effective transfer of the target style.

**Geometry-Guided Multi-View Consistency**. We conduct extensive quantitative experiments to evaluate the effectiveness of Geometry-Guided Attention (GGA) in improving multi-view consistency, with the results summarized in Tab. 3. Our method achieves consistent improvements across all metrics, with particularly notable gains in long-term consistency. We further investigate the role of the geometry-aware mask $\mathcal{M}_G$, which is designed to prevent redundant optimization over geometrically overlapping regions that could otherwise disrupt view consistency. The results show that removing $\mathcal{M}_G$ results in only marginal performance degradation, since GGA already establishes strong multi-view correspondences. This observation further highlights the effectiveness of GGA in enforcing multi-view consistency. The corresponding qualitative results are shown in Fig. 9. Without explicit multi-view modeling, local features of the original content tend to become blurred and overly smoothed. In contrast, we leverage geometric information to model cross-view relationships, preserving

sharp local details and coherent structures across different viewpoints, thereby significantly improving visual quality.

### 5.3. Limitation

During optimization, we freeze the 3DGS geometry to prevent shape collapse, which also limits the ability of our method to model geometry-level style variations. For styles with strong geometric characteristics, our method does not explicitly deform the underlying 3D structure; instead, it approximates these geometric cues as semantic textures. While this design enables the transfer of distinctive visual attributes, such as fragmented brushstrokes and unique color palettes, it may be insufficient for styles that require substantial geometric deformation. In addition, our current framework is not directly applicable to recent diffusion architectures such as SD3 and FLUX. Extending our method to these architectures may require additional adaptations or tailored optimization strategies.

## 6. Conclusion

In this work, we propose DiffStyle3D, a novel diffusion-based paradigm for 3DGS stylization that operates directly in the latent space, thereby avoiding unstable denoising guidance. It introduces an Attention-Aware Loss for style transfer and content preservation, and a Geometry-Guided Multi-View Consistency that injects geometric information into self-attention to form Geometry-Guided Attention, enabling cross-view correspondence modeling. Additionally, a geometry-aware mask enhances multi-view consistency by avoiding redundant optimization in overlapping regions. Extensive experiments demonstrate that DiffStyle3D achieves superior stylization quality, better visual quality, and improved content preservation compared to existing methods. Our method can facilitate high-quality and view-consistent 3D stylization, providing a useful foundation for controllable 3D asset editing and creative 3D content generation.

## Impact Statement

This paper presents DiffStyle3D, a method aimed at advancing 3DGS stylization by improving efficiency, stability, and multi-view consistency. The proposed approach may benefit applications such as virtual reality, gaming, digital art, and film production by lowering the cost and technical barriers for creating high-quality stylized 3D assets. As with other generative and stylization methods, there is a potential risk of misuse for producing misleading or deceptive visual content, as well as concerns related to bias or inappropriate use of artistic styles. These risks are not unique to our work and are common across generative modeling research. Overall, we believe this work contributes positively to the field of 3D vision without introducing significant new ethical concerns.

## Acknowledgment

This project was sponsored by Natural Science Foundation of Shanghai 25ZR1402138, and Shanghai Pujiang Programme 24PJD030. Changshuo Wang was supported in part by the European Union's Horizon 2024 Research and Innovation Programme for the Marie Skłodowska-Curie Actions under Grant No. 101211118.

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

## A. Implementation Details

Stable Diffusion 1.5 contains a total of 16 self-attention layers distributed across multiple resolution levels: 2 layers in the 64×64 DownBlock, 2 in the 32×32 DownBlock, 2 in the 16×16 DownBlock, 1 in the 8×8 MidBlock, 3 in the 16×16 UpBlock, 3 in the 32×32 UpBlock, and 3 in the 64×64 UpBlock. We extract all self-attention layers and use them for the computation of the Attention-Aware Loss, thereby fully accounting for information across different spatial scales. The optimization process of DiffStyle3D is conducted for 600 iterations, with learning rates set to 0.02 for the spherical harmonic basis functions and 0.000167 for the higher-order spherical harmonic functions, respectively. Adam is used as the optimizer.

## B. Metrics

FID is computed between the stylized results and the content images. We employ a pre-trained VGG19 network to extract features from both content and style layers to evaluate the fidelity of the style transfer. Specifically, content consistency is quantified by the distance between high-level feature representations, while style similarity is modeled via the correlations between feature maps (i.e., Gram matrices). Accordingly, the $S_{vgg}$ metric is defined as:

$$S_{vgg} = \frac{1}{N}\sum_{i=1}^{N}(\text{VGG}_{content}(I_i, I_i^c) + 1000 * \text{VGG}_{style}(I_i, I^s)) \tag{15}$$

We adopt the evaluation metrics used in CLIPGaussian, and the detailed definition of each metric is given as follows:

$$\text{CLIP-C} = \frac{1}{N}\sum_{i=1}^{N}\cos(E_{CLIP}(I_i), E_{CLIP}(I_i^c)) \tag{16}$$

$$\text{CLIP-S} = \frac{1}{N}\sum_{i=1}^{N}\cos(E_{CLIP}(I_i), E_{CLIP}(I^s)) \tag{17}$$

$$\text{CLIP-CONS} = \frac{1}{N-1}\sum_{i=1}^{N-1}\cos(E_{CLIP}(I_{i+1}) - E_{CLIP}(I_i), E_{CLIP}(I_{i+1}^c) - E_{CLIP}(I_i^c)) \tag{18}$$

$$\text{CLIP-F} = \frac{\sum_{i=1}^{N-1}cos(E_{CLIP}(I_{i+1}) - E_{CLIP}(I_i))}{\sum_{i=1}^{N-1}cos(E_{CLIP}(I_{i+1}^c) - E_{CLIP}(I_i^c))} \tag{19}$$

## C. Comparison Methods

We do not evaluate StyleGaussian on the object-level dataset for two reasons. First, unlike other methods, StyleGaussian introduces additional Gaussian model parameters, which prevent a straightforward substitution of camera parameters for object-level evaluation, making a fair comparison with other methods difficult. Second, StyleGaussian already demonstrates poor performance in both style transfer quality and content preservation on the scene-level dataset; therefore, we do not include it in the object-level evaluation.

## D. Experiments

**Qualitative Comparisons**. We provide additional qualitative comparisons in Fig. 10 and Fig. 11. Compared with FantasyStyle, our method produces stylized results that more closely align with the target style. In contrast, other methods often suffer from over-stylization, which leads to the loss of original content, or from content leakage originating from the style image. Our approach effectively mitigates these issues, achieving more faithful style transfer while better preserving the original content.

**More Results**. Fig. 12 shows more results produced by our method. Leveraging the rich representations of the diffusion model, our approach is able to transfer diverse and complex styles accurately while avoiding content leakage from the style images and preventing over-stylization. We provide additional video results in the Supplementary Material.

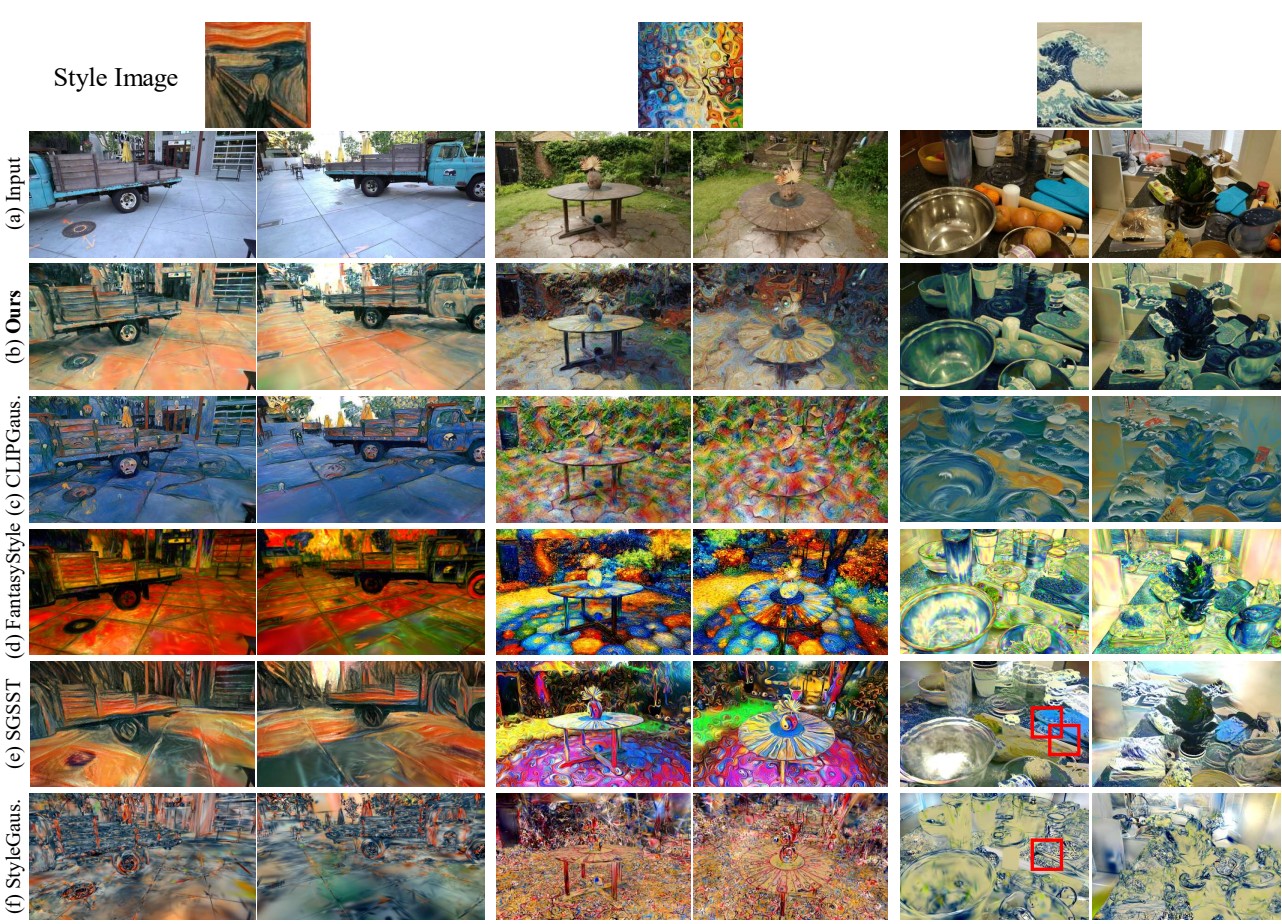

*Figure 10.* More qualitative comparison of different methods on scene-level datasets.

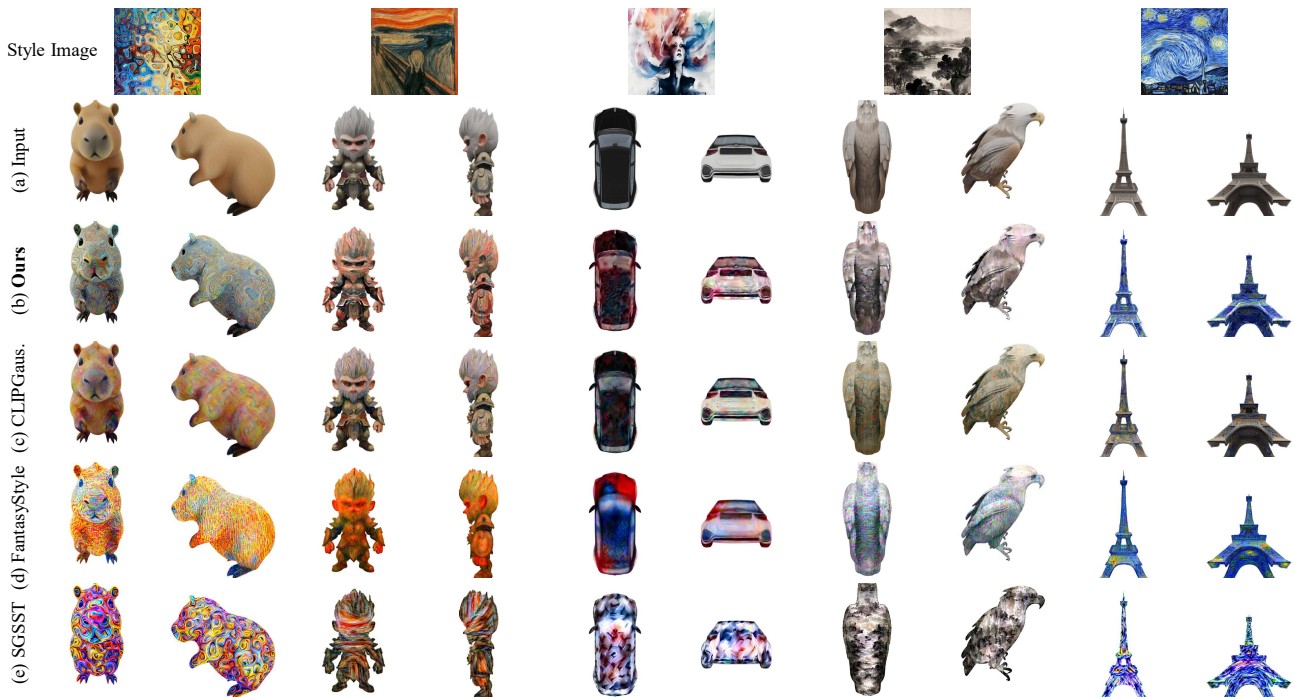

*Figure 11.* Qualitative comparison of different methods on object-level datasets. Other methods often suffer from over-stylization and content leakage from the style image. In contrast, our approach avoids these issues, achieving superior visual quality in style transfer.

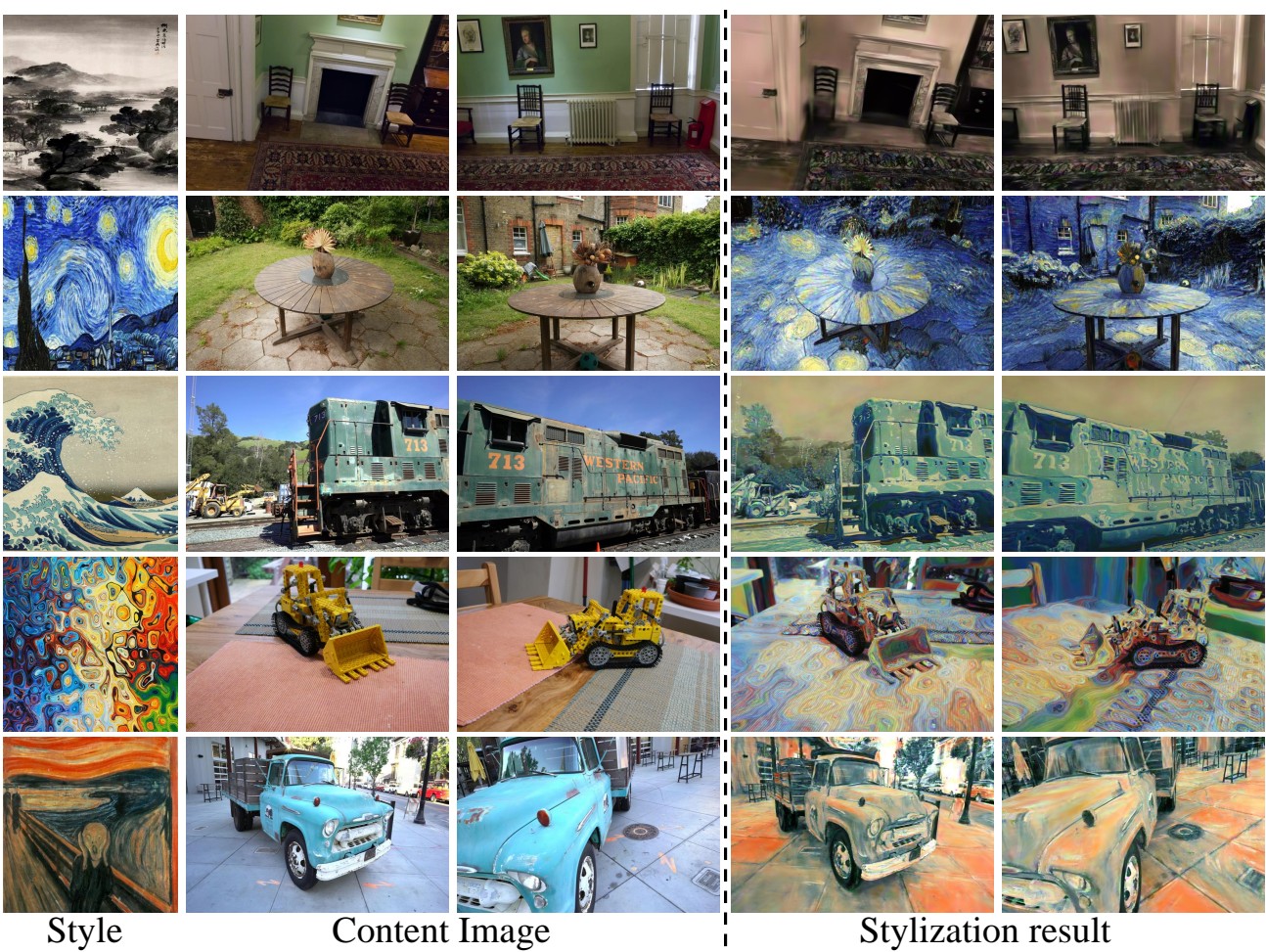

| Style | Content Image | Stylization result |
|---|---|---|

*Figure 12.* More visualization results of our method.

