# OpenReview forum: "DiffStyle3D: Consistent 3D Gaussian Stylization via Attention Optimization"
_ICML.cc/2026/Conference — ICML 2026 regular_

### Official Review · Reviewer_DVw2 · 2026-03-11

**Soundness:** 3
**Presentation:** 2
**Significance:** 3
**Originality:** 2
**Overall Recommendation:** 4
**Confidence:** 2

**Summary:**

This paper proposes a 3D Gaussian stylization method based on attention optimization. Specifically, the method introduces two types of objectives. The first objective reduces the distance between the tensor obtained after applying attention to the style-injected tensor and the tensor of the rendered images. The second objective preserves the original content by reducing the gap between the attention tensor of the original images and that of the rendered images. To enhance multi-view consistency, the method aggregates multi-view tokens based on a geometric prior, namely the depth from 3D Gaussian Splatting, when applying the attention optimization objectives.

**Compliance With Llm Reviewing Policy:**

Affirmed.

**Final Justification:**

Authors resolves my concerns, but I agree with the concerns of other reviewers. So I maintain my original scores.

**Key Questions For Authors:**

- Since both the diffusion model and the VAE appear to be involved in backpropagation, the optimization process seems potentially expensive. Does this not make training slow? If the overall training time is still short, is it because the method requires only a small number of iterations?

- In style transfer, the choice of which layer to inject the style into is often important. Have the authors conducted experiments on this design choice? If so, how does the injection layer affect the final results?

**Limitations:**

They did not mention limitations

**Strengths And Weaknesses:**

## Strengths

- The proposed method is intuitive and easy to understand, as it transfers the style of the rendered image by reducing the gap toward the style-injected tensor.

- The geometry-guided multi-view consistency design is reasonable, since it explicitly aggregates multi-view tensors and encourages them to imitate the stylized tensor.

## Weaknesses
- The paper does not include a user study. Since style transfer is inherently perceptual, a user study would help better validate the qualitative advantages of the proposed method.

- The comparison with recent baselines is limited. In particular, it would be beneficial to compare against more recent methods such as Tune-Your-Style [A].


[A] Zhao, Yian, et al. "Tune-Your-Style: Intensity-tunable 3D Style Transfer with Gaussian Splatting." Proceedings of the IEEE/CVF International Conference on Computer Vision. 2025.

---

> ### Author Rebuttal · Authors · 2026-03-30
>
> Thank you for recognizing that our method is easy to understand and that the geometry-guided multi-view consistency design is reasonable.
>
> **W1:** We conduct a user study to evaluate both **stylistic fidelity and content preservation**. A total of **730 votes are collected from 73 participants**. Each participant is shown ten random instances and asked to select the result that best meets the criteria. As shown in the voting results below, our approach significantly outperforms the baseline methods, **securing 43.29% of the overall user preference and demonstrating the effectiveness of our method.**
> |Method|Ours|StyleGaussian|SGSST|FantasyStyle|CLIPGaussian|
> |-|-|-|-|-|-|
> |Rank 1(%)↑|**43.29**|7.26|28.08|17.26|4.11|
>
> **W2:** So far, **Tune-Your-Style[A] has released its paper website but not the code repository**, making a fair comparison difficult. We will discuss this method in the Related Work section of the revised version.
> [A] Zhao, Yian, et al. "Tune-Your-Style: Intensity-tunable 3D Style Transfer with Gaussian Splatting." Proceedings of the IEEE/CVF International Conference on Computer Vision. 2025.
>
> **Q1:** Although the diffusion model 1.5 and VAE have relatively large parameter counts, **our Attention-Aware Loss, by centering and normalizing features, enables faster convergence with fewer iterations, resulting in shorter training time**, as emphasized in the right column of the main paper (L191–193) and Fig. 8. We also provide more detailed implementation details in the Appendix (L610–L612); our method achieves effective 3D style transfer with only 600 iterations.
>
> **Q2:** SD 1.5 contains a total of 16 self-attention layers distributed across multiple resolution levels: 2 layers in the 64×64 DownBlock, 2 in the 32×32 DownBlock, 2 in the 16×16 DownBlock, 1 in the 8×8 MidBlock, 3 in the 16×16 UpBlock, 3 in the 32×32 UpBlock, and 3 in the 64×64 UpBlock. **We conducted ablation studies on the choice of self-attention layers is shown in the table below.** The quantitative results indicate that extracting features in earlier layers (e.g., layer 0-7) yields stronger style fidelity (CLIP-S: 0.65) but degrades content preservation (CLIP-C: 0.70) and multi-view consistency. Conversely, utilizing later layers (e.g., layer 7-16) better preserves content and view consistency (Short-range LPIPS: 0.074) but results in weaker stylization (CLIP-S: 0.62). **By extracting and computing self-attention features across all layers (Ours), our method achieves the optimal trade-off, matching the highest style transfer performance (CLIP-S: 0.65) while maintaining strong content preservation (CLIP-C: 0.71) and multi-view consistency**.
>
> |Method|CLIP-S↑|CLIP-C↑|Short-range consistency (LPIPS↓)|Short-range consistency (RMSE↓)| Long-range consistency (LPIPS↓)| Long-range consistency (RMSE↓)|
> | -|-|-|-|-|-|-|
> |layer0-2|0.60|0.74|0.079|0.081|0.146|0.138|
> |layer0-4|0.62|0.72|0.079|0.080|0.145|0.136|
> |layer0-6|0.65|0.70|0.077|0.078|0.144|0.134|
> |layer7-16|0.62|0.73|0.074|0.074|0.139|0.129|
> |layer10-16|0.62|0.73|0.075|0.075|0.140|0.129|
> |layer13-16|0.60|0.75|0.078|0.079|0.144|0.133|
> |Ours(all)|0.65|0.71|0.075|0.075|0.141|0.130|
>
> **Limitations**: We discuss the limitations in the Impact Statement (L447–452):"*As with other generative and stylization methods, there is a potential risk of misuse for producing misleading or deceptive visual content, as well as concerns related to bias or inappropriate use of artistic styles. These risks are not unique to our work and are common across generative modeling research.*" In the revised version, we will add a dedicated “Limitations” section for a more detailed discussion.

---

> > ### Author Rebuttal · Reviewer_DVw2 · 2026-04-03
> >
> > Thanks for the authors rebuttal, especially for the user study and analysis on the layer selection.
> >
> > However, after I read the concerns from the other reviewers, I partially agree with the other reviewer's concern that the paper lacks sufficient theoretical justification for multi-view consistency when using 2D diffusion, or for the reason for using 3D diffusion was not adopted, and therefore I will keep my score weak accept.

---

> > > ### Author Response · Authors · 2026-04-03
> > >
> > > Thank you for sharing your concerns and your positive score.
> > >
> > > Many existing methods[1,2,3,4] leverage self-attention to enforce consistency across different views. For example, TokenFlow[1] demonstrates that tokens in the self-attention layers of 2D diffusion models exhibit strong temporal and spatial correspondences, which can be explicitly propagated to enforce consistency. Text2Video-Zero[2] requires no fine-tuning on video data; by introducing cross-frame self-attention based on self-attention into existing 2D diffusion models, it achieves strong temporal consistency in generated videos. **We propose a Geometry-Guided Attention mechanism that incorporates geometric information into self-attention to enhance multi-view consistency, which is the key innovation of our method. As shown in Tab. 2, Geometry-Guided Attention effectively improves consistency.**
> > >
> > > [1] TokenFlow: Consistent Diffusion Features for Consistent Video Editing, ICLR 2024
> > >
> > > [2] Text2Video-Zero: Text-to-Image Diffusion Models are Zero-Shot Video Generators, ICCV 2023
> > >
> > > [3] MVDiffusion: Enabling Holistic Multi-view Image Generation with Correspondence-Aware Diffusion, NeurIPS 2023
> > >
> > > [4] SyncDreamer: Generating Multiview-consistent Images from a Single-view Image, ICLR 2024
> > >
> > > **More importantly, style images are inherently 2D. While 3D models offer strong multi-view consistency, they struggle to perform effective style transfer.** Combining 2D and 3D models is also non-trivial, as their **feature spaces are misaligned** and such integration incurs **additional computational cost**. Considering these factors, we adopt a 2D diffusion model, which enables effective style transfer while maintaining multi-view consistency.
> > >
> > > Thank you for your time.
> > >
> > > Best regards,
> > >
> > > Authors of Paper #7011

---

### Official Review · Reviewer_6Fg3 · 2026-03-13

**Soundness:** 3
**Presentation:** 3
**Significance:** 2
**Originality:** 2
**Overall Recommendation:** 3
**Confidence:** 4

**Summary:**

This paper proposes DiffStyle3D, a diffusion-based paradigm for 3D Gaussian Splatting (3DGS) style transfer. DiffStyle3D directly optimizes in the latent space of a diffusion model. Two key techniques are introduced: (1) an Attention-Aware Loss that injects style image keys (K) and values (V) into the self-attention computation of rendered content queries (Q), with centering and normalization to focus on feature directions rather than magnitudes; and (2) Geometry-Guided Multi-View Consistency (GGA), which warps attention features across views using known camera parameters and depth maps, supplemented by a geometry-aware mask to avoid redundant optimization in overlapping regions.

**Compliance With Llm Reviewing Policy:**

Affirmed.

**Key Questions For Authors:**

1. Can you provide a user study comparing perceptual quality across methods?
2. Does the method work with newer diffusion architectures (SDXL, SD3, Flux/DiT)?
3. Why does t=1 work best? Can you provide intuition or analysis beyond the empirical observation in Fig. 4? Does this hold for different style types (abstract vs. realistic)?
4. How does the method handle styles with strong geometric characteristics (e.g., cubism, low-poly)?

**Limitations:**

They do not discuss technical limitations such as the restriction to color-only stylization, dependence on SD model, the fixed timestep assumption, or computational scaling with scene complexity. A more thorough limitations section is needed.

**Strengths And Weaknesses:**

Strengths
1. The Attention-Aware Loss formulation (Eqs. 4–8) is a clean and principled way to transfer style through self-attention manipulation in latent space, avoiding the instability of denoising-direction-based approaches. The insight of injecting style K/V while keeping content Q, combined with centering and normalization, is well- motivated (Fig. 3, 8).
2. The Geometry-Guided Attention (Eqs. 9–12) elegantly leverages known 3DGS geometry to enforce cross-view consistency within the diffusion model's own attention mechanism, rather than relying on external consistency losses. The visibility mask (Eq. 10) correctly handles occlusions.

Weaknesses
1. The evaluation dataset is small: only 8 scenes × 14 styles for scene- level and 10 objects × 14 styles for object-level. No user study is provided, which is critical for style transfer where perceptual quality often diverges from automated metrics.
2. The method fixes the diffusion timestep at t=1 (Sec. 4.1), which is justified empirically (Fig. 4) but lacks theoretical grounding. Why does the smallest timestep yield the best results? Is this consistent across different diffusion models or specific to SD 1.5?
3. The method is built on Stable Diffusion 1.5, which is now outdated. It is unclear whether the approach generalizes to newer architectures (SDXL, SD3, Flux) that use different attention mechanisms (e.g., joint attention in DiT-based models).
4. The idea of manipulating self-attention K/V for style transfer has been explored in 2D (e.g., StyleAligned, MasaCtrl). The extension to 3D via geometry-guided warping is novel but incremental. The geometry-aware mask M_G shows only marginal quantitative improvement.

---

> ### Author Rebuttal · Authors · 2026-03-30
>
> Thank you for recognizing that our method is well-motivated.
>
> **W1&Q1**: **StyleGaussian evaluates 9 scenes**, but its stylization quality is poor(see Fig.5), limiting practical use. **SGSST performs 40 groups** of 3D style transfer experiments across 9 scenes with different style images. **FantasyStyle applies 8 styles to 17 scenes(136 cases)**, while **CLIPGaussian evaluates 2 scenes and 2 objects with 4 prompts and 4 style images(32 cases)**. In contrast, our dataset includes 8 scenes and 10 objects, each paired with 14 style images, **totaling 252 cases**. This scale is substantially larger than prior methods and enables a more comprehensive evaluation. We collected **730 votes from a total of 73 participants** to evaluate performance in terms of style transfer and content preservation. User study is shown in the table below. It is clearly observed that our method outperforms the others.
> |Method|Ours|StyleGaussian|SGSST|FantasyStyle|CLIPGaussian|
> |-|-|-|-|-|-|
> |Rank 1(%)↑|**43.29**|7.26|28.08|17.26|4.11|
>
> **W2&Q3:** **The choice of fixing t=1 stems from the behavior of features under different noise levels, which has been extensively studied in prior work[1,2,3] rather than being merely empirically justified**. At large timesteps, the forward diffusion process $q(x_t|x_0) = \mathcal{N}(x_t; \sqrt{\bar{\alpha}_t}x_0, (1-\bar{\alpha}_t)I)$ severely corrupts the latent representation with Gaussian noise, causing the UNet’s attention features to lose fine-grained details. Consequently, performing feature matching at **large timesteps introduces severe noise and artifacts**, as observed in Fig. 4. In contrast, when $t=1$, the noise level is minimal, allowing the UNet to function as a discriminative dense feature extractor. The resulting attention features accurately **preserve intricate details such as brushstrokes**, ensuring reliable optimization for style transfer. Importantly, this phenomenon is not unique to SD 1.5 but is also present in newer models like SDXL and SD3. It is a general property of diffusion models due to the shared nature of the forward diffusion process. **In the Appendix, Fig. 10 and 11 show the abstract style images.**
>
> [1]SDEdit: Guided Image Synthesis and Editing with Stochastic Differential Equations,ICLR2022
>
> [2]Emergent Correspondence from Image Diffusion,NeurIPS2023.
>
> [3]Prompt-to-Prompt Image Editing with Cross Attention Control,ICLR2023
>
> **W3&Q2:** We adopt **SD 1.5 as a trade-off between optimization stability and computational efficiency**. Replacing it with SDXL leads to severe instability, mainly due to **a known limitation where its VAE suffers from numerical overflow in FP16, introducing noise and artifacts during 3D optimization**. Addressing this requires FP32 VAE or heavily modified pipelines, both of which significantly increase computational cost. More recent models such as **SD3 and Flux use joint attention mechanisms that differ fundamentally from the U-Net**, requiring substantial redesign to integrate. More importantly, their computational cost **makes it infeasible to maintain a reasonable batch size (batch_size=4) on a single 48GB GPU**.
>
> **W4:** **These 2D methods and their derived 3D style transfer approaches typically rely on injecting K/V features at specific timesteps to obtain predicted noise** for optimizing 3D results (often based on score distillation and its variants). However, **this paradigm is prone to training instability and over-saturation issues**. **In contrast, our method completely eliminates the dependence on noise prediction and diffusion process simulation, introducing a novel paradigm that directly optimizes in the self-attention feature space.** This paradigm shift significantly improves training stability and avoids over-saturation, representing more than a simple incremental improvement. $M_G$ prevents repeated optimization of overlapping regions across views, which leads to inconsistencies. **Its marginal quantitative improvement indicates that Geometry-Guided Attention already achieves strong multi-view consistency, largely mitigating the conflicts that $M_G$ is intended to address.**
>
> **Q4:** During optimization, we freeze 3DGS geometry to prevent shape collapse. As a result, for styles with strong geometric characteristics, **our method does not deform the 3D geometry but instead treats it as semantic texture, enabling accurate transfer of distinctive attributes such as fragmented brushstrokes and unique color palettes**. Future work will explore depth-conditioned guidance and local rigidity constraints for geometric style transfer.
>
> **Limitations**: The limitations of color-only stylization are discussed in Q4, the dependence on the SD model in W3&Q2, and the fixed timestep assumption in W2&Q3. Input images are resized to 512×512 to match the input requirements of SD 1.5. Therefore, the computational cost does not increase with scene complexity. In the revised version, we will add a dedicated limitations section.

---

### Official Review · Reviewer_Lf7K · 2026-03-26

**Soundness:** 2
**Presentation:** 2
**Significance:** 2
**Originality:** 2
**Overall Recommendation:** 4
**Confidence:** 4

**Summary:**

This paper focuses on 3D stylization of Gaussian, and introduces an attention-based methods to supervise in latent space for effectiveness and efficiency. To achieve this goal, this paper propose an attention-aware loss and geometry-guided multi-view consistency to preserve content information and 3D consistency.

**Compliance With Llm Reviewing Policy:**

Affirmed.

**Final Justification:**

I have read the original paper, other reviewers' comments and the authors' rebuttal. I raise my score to weak accept. However, I recommend the authors to further discuss the following questions in their final version.

1. Application of image diffusion models. There is no ablation study on the effectiveness of the diffusion model. It is still confusing for me whether the mv-consistency is from self-attention or GGA. The demonstration of the mv-consistency of self-attention is weak. Either video generation model or 3D diffusion models trained on mv images are cited. I guess that replacing the  SD 1.5 by a mv diffusion model leads to a better performance.

2. Removing GGA leads to only marginal performance degradation. Removing the geometry-aware mask has almost no measurable impact at all.

3. Clarification with 2D style transfer. While the attention mechanism has already been discussed in [1], including both content and structure. The contribution of this method should be further discussed.
[1] Cross-Image Attention for Zero-Shot Appearance Transfer, SIGGRAPH 2024

**Key Questions For Authors:**

Please refer to weakness.

**Limitations:**

I recommend the authors to discuss about the application of 3D stylization, which is not obvious to me. It is hard to figure out the meaning of generated 3D scene with such texture in Figure 5,6,7. This discussion is helpful to point out a clear future path for this topic to truely benefit our work and life.

**Strengths And Weaknesses:**

Strength:
1. This paper proposes an attention-based strategy to align style image, content images and render images for style transfer.
2. This paper considers the multi-view consistency and introduce a GGA module as well a geometry-aware mask.
3. The experiments demonstrate the effectiveness of this attention-based mechanism.
Weakness:
1. Motivation on Diffusion-based paradigm. The authors claim that directly optimizing in the latent space is superior to following denoising directions. As illustrated in Fig. 4, the diffusion model is used as a frozen feature extractor, especially at t=1. I wonder why is a complex diffusion model necessary if its generative power is not used.
2. Geometry-Guide Attention. The authors claim that the geometry-guided attention can strengthen cross-view consistency. Why does the content image not pass this GGA module but a classic SA module？
3. Ablation on Content Loss. I did not find any ablation study on the content loss. Its effectiveness is not clear. Since only color-related parameters are optimized, the original geometry is fixed. The structure and content seem to be stable and maintained. The claimed "over-stylization" may be simply solved by less training iteration or smaller learning rate.
4. The selection of key and values. The authors claim that directly aligning the attention outputs of the style image and the rendered image leads to content transfer. Why can replacing the query from style image to the rendered image address this issue?
Minor weakness
1. The formulas are quite confusing and there are so many symbols, variables in this paper. For example, the projection process may be presented more clearly instead of the equation 9. Besides, the W_{b<-j} is also confusing, which is defined as "the bilinear warping operator guided by g". Also, I guess D_b{p} represents the depth of p, which is not defined in the paper. Therefore, I sincerely recommend the authors to reorganize the method part to simplify the presentation with less new symbols and more explanation sentences.

---

> ### Author Rebuttal · Authors · 2026-03-30
>
> Thank you for recognizing the effectiveness of this attention-based mechanism.
>
> **W4**: **Diffusion models are trained on billions of text–image pairs for high-quality image synthesis, enabling their feature space to implicitly encode rich generative priors.** They produce fine-grained spatial correspondences alongside rich semantic representations.
> Compared to traditional feature extractors, VGG primarily captures low-level textures but lacks high-level semantic understanding, while CLIP provides strong semantic representations at the cost of dense spatial resolution. Diffusion models clearly outperform both in this regard. **More importantly, the self-attention mechanism in diffusion models is particularly well suited for enforcing multi-view consistency, a critical capability that other feature extractors fundamentally lack.**
>
> **W5**: **The content images are derived from the original 3D scene and inherently exhibit multi-view consistency.** Since they remain unchanged throughout the optimization process, there is no need to enforce this property using the GGA module. In contrast, rendered images tend to lose multi-view consistency during stylization, and therefore require GGA.
>
> **W6**：Ablation on content loss (λ) is presented in the table below. Appropriately increasing λ helps mitigate over-stylization while enhancing content preservation. **Although our method keeps the 3D geometry fixed, unconstrained color optimization can still blur important semantic boundaries and local textures on the surface**. For example, local details may be merged with the background into a uniform stylized color, making them indistinguishable. While reducing the number of training iterations or lowering the learning rate can alleviate over-stylization in certain cases, the complexity varies significantly across different 3D scenes and style images, **often requiring case-by-case tuning of iterations and learning rates. Such per-scene hyperparameter adjustment is costly and limits the practicality of the method in real-world applications**. Furthermore, applying scene-specific fine-tuned parameters across the evaluation dataset would **lead to unfair comparisons with baseline methods.**
> |Method|CLIP-S↑|CLIP-C↑|FID↓|
> |-|-|-|-|
> |λ=0|0.70|0.67|267.9|
> |λ=0.5|0.57|0.81|97.7|
> |λ=1.0|0.49|0.92|43.3|
> |Ours(λ=0.1)|0.65|0.71|204.8|
>
> **W7**：If the attention outputs of the rendered image and the style image are directly aligned, it essentially forces them to become identical in the feature space. This leads the rendered image to excessively replicate the structure and content of the style image, resulting in significant content leakage (as shown in the third column of Fig.3). **In the self-attention mechanism, $Q$ primarily determines the spatial layout and structural topology of the image; $K$ serves as a semantic matching index for retrieving relevant features; and V encodes the actual visual appearance, including texture, color, and overall style attributes.** Therefore, we use the $K$ and $V$ from the style image together with the $Q$ from the rendered image to perform attention computation. This design preserves the original structure of the rendered image while introducing the appearance characteristics of the style image, effectively preventing content leakage and enabling more natural and stable style transfer. **This strategy has been similarly observed in StyleAligned[1] and related works[2][3].**
>
> [1] Style aligned image generation via shared attention, CVPR 2024
>
> [2] Cross-Image Attention for Zero-Shot Appearance Transfer, SIGGRAPH 2024
>
> [3] Style Injection in Diffusion: A Training-free Approach for Adapting Large-scale Diffusion Models for Style Transfer, CVPR 2024
>
> **Minor-W8**: The warping operator $\mathcal{W}_{b \leftarrow j}$ represents a standard differentiable grid sampling (bilinear interpolation) guided by $\mathbf{g}{\scriptstyle b \leftarrow j}$, and $D_b(\mathbf{p})$ denotes the depth value at pixel $\mathbf{p}$ in view $b$, as respectively stated in L235-236 and L238-239 of the left column of the main paper. We will provide more detailed descriptions for these symbols in the revised manuscript.
>
> **Limitations**: In L41–50 of the Introduction and L442–447 of the Impact Statement, we have already discussed the applications of 3D stylization. Our method will benefit applications in virtual reality, gaming, digital art, and film production by lowering the cost and technical barriers to creating high-quality 3D assets with diverse artistic styles. We will further elaborate on this in the Conclusion section of the revised version.

---

> > ### Author Rebuttal · Reviewer_Lf7K · 2026-04-03
> >
> > I do appreciate the authors' rebuttal, which has solved part of my issues. Futher concerns based on the rebuttal are listed as follows.
> > W4: In the rebuttal, the authors claim that the self-attention mechanism in diffusion models is particularly well suited for enforcing multi-view consistency. However, those diffusion models, especially image diffusion models, are not trained on multi-view images. Why are they well suited for enforcing multi-view consistency? 3D diffusion models possess such ability, but the experiments are based on Stable Diffusion 1.5, which does not introduce any 3D training data for multi-view consistency. Besides, the chosen of t=1 (totally about 1,000 timesteps) also limits its generative power restrictly.
> > W6: The ablation study indicates that the results are senstive to $\lambda$. I have two concerns on this ablation study. First, with larger $\lambda$, the model achieves better performance on CLIP-C and FID. Why is the $\lambda$ set to 0.1. What is the relation between the matrix and the performance? Does better CLIP-C and FID indicate the over-stylization? Second, the authors claim that adjusting learning rate and iteration requires case by case tuning. I wonder if fixed learning rate and iteration can avoid this problem? According to the ablation study, the results are also senstive to $\lambda$. Why is the $\lambda$ fixed instead of case by case tuning. The superiority of the content loss is not persuasive.

---

> > > ### Author Response · Authors · 2026-04-03
> > >
> > > Thank you for letting us know your concern.
> > >
> > > **W4**: Many existing methods[1,2,3,4] leverage self-attention to enforce consistency across different views. For example, TokenFlow[1] demonstrates that tokens in the self-attention layers of 2D diffusion models exhibit strong temporal and spatial correspondences, which can be explicitly propagated to enforce consistency. Text2Video-Zero[2] requires no fine-tuning on video data; by introducing cross-frame self-attention based on self-attention into existing 2D diffusion models, it achieves strong temporal consistency in generated videos. **We propose a Geometry-Guided Attention based on self-attention to enhance multi-view consistency, which is the key innovation of our method.**
> > >
> > > **Since our goal is style transfer, the style information can already be effectively applied at t=1, and additional generative capacity from the diffusion process is unnecessary.** Moreover, as shown in Fig. 4, larger timesteps introduce more noise, while intermediate timesteps do not yield additional details.
> > >
> > > [1] TokenFlow: Consistent Diffusion Features for Consistent Video Editing, ICLR 2024
> > >
> > > [2] Text2Video-Zero: Text-to-Image Diffusion Models are Zero-Shot Video Generators, ICCV 2023
> > >
> > > [3] MVDiffusion: Enabling Holistic Multi-view Image Generation with Correspondence-Aware Diffusion, NeurIPS 2023
> > >
> > > [4] SyncDreamer: Generating Multiview-consistent Images from a Single-view Image, ICLR 2024
> > >
> > >
> > > **W6**: As noted in the main paper (left column, L315–316), **CLIP-S is used to assess the quality of style transfer** between the stylized image and the reference style image, while **FID and CLIP-C measure content preservation** between the original content image and the stylized image. Therefore, **an excessively large λ leads to incomplete stylization, resulting in higher CLIP-C and FID scores.** The ablation results for λ are shown in the table below. Through extensive experiments, **we find that λ=0.1 provides a good balance between stylization and content preservation.** When λ=0, stylization is significantly enhanced, but content preservation deteriorates sharply (reflected by a large increase in FID). In contrast, when λ=0.5, stylization becomes insufficient, leading to a noticeable decrease in FID.We fix the learning rate, number of iterations, and λ to ensure a fair comparison with other methods.
> > > **In practical applications, our parameter settings generalize well across most scenarios, as demonstrated in Tab. 1 and Fig. 5, 6. In certain cases, users can also dynamically adjust the parameters according to their specific needs.**
> > > |Method|CLIP-S↑|CLIP-C↑|FID↓|
> > > |-|-|-|-|
> > > |λ=0|0.70|0.67|267.9|
> > > |λ=0.5|0.57|0.81|97.7|
> > > |λ=1.0|0.49|0.92|43.3|
> > > |Ours(λ=0.1)|0.65|0.71|204.8|
> > >
> > > If our responses have addressed your concerns, we would greatly appreciate your consideration in updating your score to Borderline Accept (4). Your support would mean a lot to us and strongly encourage our continued work in this direction.
> > >
> > > Thank you for your time.
> > >
> > > Best regards,
> > >
> > > Authors of Paper #7011

---

### Decision · Program_Chairs · 2026-04-30

**Decision:**

Accept (regular)

**Comment:**

This paper received 2 weak accepts and 1 weak reject. The reviewers agree that this paper is well-motivated and with solid experiments. The rebuttal successfully addressed the concerns of the reviewers. As the reviewer who gave the only negative score didn't reply to the rebuttal, the AC checked the comments and found in general the weaknesses have been resolved. Therefore the AC recommends acceptance.